# Noise Cancellation of Helicopter Blade Deformations Measurement by Fiber Bragg Gratings

**DOI:** 10.3390/s21124028

**Published:** 2021-06-11

**Authors:** Raoul R. Nigmatullin, Timur Agliullin, Sergey Mikhailov, Oleg Morozov, Airat Sakhabutdinov, Maxim Ledyankin, Kamil Karimov

**Affiliations:** 1Department of Radio-Electronic and Information-Measuring Tools, Kazan National Research Technical University Named after A.N. Tupolev-KAI, K. Marx Str. 10, 420111 Kazan, Russia; 2Department of Radiophotonics and Microwave Technologies, Kazan National Research Technical University Named after A.N. Tupolev-KAI, K. Marx Str. 10, 420111 Kazan, Russia; taagliullin@mail.ru (T.A.); microoil@mail.ru (O.M.); azhsakhabutdinov@kai.ru (A.S.); mail12kamil2000@mail.ru (K.K.); 3Department of Aero-Hydrodynamics, Kazan National Research Technical University Named after A.N. Tupolev-KAI, K. Marx Str. 10, 420111 Kazan, Russia; samikhailov@kai.ru (S.M.); maledyankin@kai.ru (M.L.)

**Keywords:** fiber Bragg grating, noise cancellation, helicopter blade deformation measurement, procedure of the optimal linear smoothing, 3D-invariant method

## Abstract

The work presents data treatment methods aimed at eliminating the noise in the strain sensor data induced by vibrations of the helicopter blade in flight conditions. The methods can be applied in order to enhance the metrological performance of the helicopter weight estimation system based on the deformation measurement of the main rotor blades. The experimental setup included a composite plate fixed to the vibrating stand on the one end, with six fiber-optic strain sensors attached to its surface. In this work, the procedure of the optimal linear smoothing (POLS) and 3D-invariant methods were used to obtain monotone calibration curves for each detector, thereby making it possible to distinguish the increase of load applied to the free end of the plate with an increment of 10 g. The second method associated with 3D invariants took into account 13 quantitative parameters defined as the combination of different moments and their intercorrelations up to the fourth-order inclusive. These 13 parameters allowed the calculation of the 3D surface that can serve as a specific fingerprint, differentiating one set of initial data from another one. The combination of the two data treatment methods used in this work can be applied successfully in a wide variety of applications.

## 1. Introduction

Helicopter main rotor blades are essential components that define the flight performance and safety of the helicopter. Therefore, monitoring of their state is of the utmost importance during helicopter exploitation. Moreover, a sophisticated aeroelastic analysis of the rotor blades is necessary at the early stages of helicopter development, since the blades are subjected to significant displacements during flight and the aerodynamic conditions around the rotor are highly unsteady and complex [1]. In order to perform the abovementioned tasks, real-time blade deformation measurements are required, which enable the blade wear or damage detection during helicopter operation as well as the evaluation of blade displacements during development testing procedures. 

The existing approaches to helicopter blade deformation measurement can be divided into four main categories: the systems based on the detection of reflected light [2,3,4], the systems based on laser units [5], the systems based on electrical resistance strain gauges [6,7,8], and the fiber-optic strain measurement systems [9,10,11].

The methods of the first group capture the light reflected from the rotor blades in order to determine their deformation. Thus, the system presented in [2] uses light from the external source that is modulated by the rotating blades to define the amplitude of blade flapping, its velocity, bending, etc. The system described in [3] comprises CCD detectors installed in the helicopter fuselage and is able to determine the blade pitch angle and flapping. The work [4] presents an approach to blade deformation monitoring based on the digital image correlation (DIC) technique, in which the surface of the blade is painted with a high-contrast random dot pattern. Two high-resolution digital cameras are separated by a certain distance and view the same area on the blade so that the parallax is used to measure three-dimensional displacements of the blade surface. In general, the main advantage of the systems based on the reflected light detection is that they do not require the structural modification of the blade. On the other hand, the performance of such systems depends on the conditions of the ambient environment, such as weather conditions, altitude, time of day, disturbances from external light sources.

In order to increase the measurement accuracy of blade deformation, in comparison with the aforesaid methods, systems based on laser units have been proposed. Thus, in [5], Projection Moiré Interferometry (PMI) with infrared laser source is applied to measure dynamic rotor blade deflections. Similarly, to the previous group of approaches, the laser units-based methods are realized without blade structural modifications, while not being affected by disturbances from external visible light sources. However, the accuracy of such approaches also depends on the weather conditions and optical density of the environment.

Electrical resistance strain gauges have been applied for the development of rotor blade strain measurement systems for more than four decades [6]. The NASA/Army UH-60A Airloads Program was completed in 1994 [7,8] and performed over 200 flights, collecting data from the instrumentation system installed on the main rotor of the helicopter. The system included 25 strain gauges, accelerometers, pressure transducers, temperature sensors, and acquired a vast amount of data, including deformation to estimate the airloads. Unlike the methods that detect optical radiation reflected from the blades, the systems based on electrical resistance strain gauges are immune to the influence of weather conditions and generally provide higher accuracy and resolution of measurements. However, the implementation of electrical resistance strain gauges requires the corresponding modifications of the blade structure. In addition, despite the low weight and compact dimensions of the gauges, the installation of a high number of such sensors may lead to a significant increase in the overall blade weight due to the metal wires connecting the gauges.

At present, a considerable amount of research activity is aimed at the development of fiber-optic sensor systems for blade deformation measurement. The main advantages of FBG-based techniques are low weight, immunity to electromagnetic interference, small dimensions (the optical fiber diameter is 125 μm), and the absence of an electrical power supply to the sensors [9]. Moreover, optical fibers can be embedded into composite structures during fabrication [10] or attached to the surface of the blade [11]. The experimental setup presented in [12] includes 60 fiber-optic sensors based on fiber Bragg gratings (FBG) nonuniformly allocated along four optical fibers so that one-third of the sensors are positioned close to the blade root where the strain gradients are higher. The system is used to measure blade deformation in vertical and lateral directions. The system discussed in [13] uses 16 FBGs interrogated by optical frequency domain reflectometry (OFDR) and is capable of retrieving the first and second bending modes of the blade.

The application of blade deformation measurements to aeroelastic analysis requires the detection of blade oscillatory deformations during aerodynamic testing procedures. However, an alternative application for such systems can be found in the vertical displacement definition of the blades in order to estimate the gross vehicle weight of a helicopter. The development of such systems significantly contributes to the safety of helicopters, since, at present, the weight is generally determined by manually tracking the weight of cargo, passengers, fuel, etc., which does not guarantee adequate accuracy. The work [14] proposes a system that calculates the weight of a helicopter by measuring and averaging the distances between the fuselage and the underside of each blade of the main rotor while in hover or level unaccelerated forward flight. The system comprises two laser distance measuring devices at the front and rear of the fuselage. Predetermined calibration data are used to compensate for the effects of vertical and horizontal drag forces on the fuselage due to the spinning rotor and forward velocity. The task of blade vertical displacement measurement can be also accomplished using strain sensors installed on the blades [14], similarly to the previously mentioned developments. In this application, the oscillatory behavior of the blade vertical displacements caused by aeroelastic effects may deteriorate the accuracy and resolution of the weight measurement. The problem is illustrated in Figure 1, in which Figure 1a represents a qualitative graph of a signal from the strain sensor located on a steady blade under the load that increased over time in five fixed steps, while Figure 1b shows the qualitative graph of the signal from the same sensor on the oscillating blade under the same sequence of loads. As it can be seen from this example, the oscillations significantly complicate the task of distinguishing the loads acting on the blade.

The aim of the current work is to propose the method for the elimination of noise in the strain measurement data caused by the blade oscillations, thereby enhancing the metrological performance of the helicopter weight measurement system. 

## 2. Deformation Measurement Using Fiber Bragg Gratings

A fiber Bragg grating (FBG) is a segment of an optical fiber that reflects particular wavelengths of optical radiation and transmits the others. FBGs are manufactured by inscribing a systematic variation of the refractive index of the fiber core. The central wavelength of the reflected light is called the Bragg wavelength and is defined according to the following generally accepted formulation:(1)$$\lambda_{B}=2n_{eff}\Lambda ,$$
where *n_eff_* is the effective refractive index of the grating, and Λ is the grating period, i.e., the period of the refractive index variation.

The working principle of FBG is based on the change of either the effective refractive index (*n_eff_*) or the grating period (Λ) and the resulting shift of its central wavelength (*λ_B_*) when the sensor is subjected to physical fields, such as strain, temperature, pressure, humidity, etc. [15]. Figure 2a shows the schematic representation of an FBG, and Figure 2b presents the typical spectral response of an FBG for transmitted and reflected light.

The most common techniques for FBG multiplexing and interrogation, such as wavelength [16], time [17], frequency [18], polarization [19], and spatial [20] division are implemented using complex electro-optical devices, including spectrum analyzers, tunable Fabry–Perot interferometers, diffraction gratings, etc. The common disadvantage of the mentioned interrogation techniques is the complexity and high cost of devices required for their implementation. In order to simplify the FBG interrogation, the usage of addressed fiber Bragg structures (AFBSs) has been proposed [21,22]. An AFBS is a type of FBG, the spectral response of which has two ultra-narrowband components. The frequency spacing between the components is called the address frequency of the AFBS, it is in the microwave range and unique for each sensor in the system, and does not change when the AFBS is subjected to strain or temperature variations. The microwave-photonic approach for AFBS interrogation utilizes a broadband light source, an optical filter with predefined linear slope spectral response, and a photodetector. The central wavelength shift of AFBS is defined using the amplitude of the beating signal generated at the photodetector at the address frequencies of the sensors. This approach is further developed with the introduction of multiaddressed fiber Bragg structures (MAFBSs) [23], which have three or more ultra-narrowband components in their spectral response. The combination of address frequencies expands the sensor capacity of the measurement system and increases the accuracy of the MAFBS central wavelength definition.

The FBG interrogation device used in the current work implemented conventional wavelength-division multiplexing of FBG sensors, where each grating had a different Bragg wavelength so that their spectral responses did not superimpose on each other during measurements. The device was based on the Ibsen^®^ I-MON spectrum analyzer and was operated by custom software. 

## 3. Noise Cancellation Methods

### 3.1. Description of the POLS Method

In order to decrease the noise evoked by vibro-oscillations, we use the procedure of the optimal linear smoothing (POLS) that was successfully applied earlier in papers [24,25,26,27,28,29] covering different regions of physics.

The basic formula is given by the following expression:(2)$$Ysm_{j}\left(w\right)=\frac{\sum_{i=1}^{N}K\left(\frac{x_{j}-x_{i}}{w}\right)y_{i}}{\sum_{i=1}^{N}K\left(\frac{x_{j}-x_{i}}{w}\right)},\;K(t)=\exp\left(-t^{2}\right),\;j=1,2,\ldots ,N$$

Here, *K*(*t*) is a smoothing kernel given in the form of the Gaussian function. The value *w* determines the width of the smoothing window. We want to remark the following attractive features of this expression:(a)This expression is linear with respect to the function *y_i_* subjected to the smoothing procedure. Therefore, it does not contain any additional distortions evoked by the possible treatment of a function *y_i_*;(b)If w >> 1, then (as it is easily seen from Expression (2)) the smoothed function *Ysm_j_* coincides with its mean value;(c)If *w* becomes close to the zero value, then the smoothing kernel *K*(*t*) coincides practically with delta function δ(*x_j_* − *x_i_*) and therefore, in this case, *Ysm_j_* (0) ≅ *y_j_*.

These attractive features allow the selection of high-frequency oscillations and leave only low-frequency “trend” that is proved to be useful for practical applications. 

In this paper, we apply the simple Expression (2) for the elimination of high-frequency oscillations, and the remaining trend allows us to select the desired parameters that differentiate the oscillations without load from oscillations with a load. 

### 3.2. Description of 3D-DGI Method 

In this section, we describe the mathematical details associated with the derivation of the complete discrete geometrical invariant (DGI) in 3D space. We remind here that preliminary results based on the application of the incomplete DGI form of the fourth order in 3D space are outlined recently in [29]. Let us consider the complete form of the fourth-order power law.
(3)$$\begin{array}{l} L_{k}^{\left(4\right)}=A_{40}^{\left(1,0\right)}{\left(y_{1}-r_{1k}\right)}^{4}+A_{40}^{\left(2,0\right)}{\left(y_{2}-r_{2k}\right)}^{4}+A_{40}^{\left(3,0\right)}{\left(y_{3}-r_{3k}\right)}^{4}-\\ -B_{22}^{\left(1,2\right)}{\left(y_{1}-r_{1k}\right)}^{2}\cdot {\left(y_{2}-r_{2k}\right)}^{2}-B_{22}^{\left(1,3\right)}{\left(y_{1}-r_{1k}\right)}^{2}\cdot {\left(y_{3}-r_{3k}\right)}^{2}\\ -B_{22}^{\left(2,3\right)}{\left(y_{2}-r_{2k}\right)}^{2}\cdot {\left(y_{3}-r_{3k}\right)}^{2}+C_{211}^{1(2,3)}{\left(y_{1}-r_{1k}\right)}^{2}\cdot \left(y_{2}-r_{2k}\right)\cdot \left(y_{3}-r_{3k}\right)\\ +C_{211}^{2(1,3)}{\left(y_{2}-r_{2k}\right)}^{2}\cdot \left(y_{1}-r_{1k}\right)\cdot \left(y_{3}-r_{3k}\right)+\\ +C_{211}^{3(1,2)}{\left(y_{3}-r_{3k}\right)}^{2}\cdot \left(y_{1}-r_{1k}\right)\cdot \left(y_{2}-r_{2k}\right)\\ -\frac{1}{2}D_{31}^{\left(1,2\right)}\left(y_{1}-r_{1k}\right)\cdot \left(y_{2}-r_{2k}\right)\left[{\left(y_{1}-r_{1k}\right)}^{2}+{\left(y_{2}-r_{2k}\right)}^{2}\right]-\\ -\frac{1}{2}D_{31}^{\left(1,3\right)}\left(y_{1}-r_{1k}\right)\cdot \left(y_{3}-r_{3k}\right)\left[{\left(y_{1}-r_{1k}\right)}^{2}+{\left(y_{3}-r_{3k}\right)}^{2}\right]-\\ -\frac{1}{2}D_{31}^{\left(2,3\right)}\left(y_{2}-r_{2k}\right)\cdot \left(y_{3}-r_{3k}\right)\left[{\left(y_{2}-r_{2k}\right)}^{2}+{\left(y_{3}-r_{3k}\right)}^{2}\right]. \end{array}$$

In Expression (3), the upper indices define the combination of the variables *y*_α_ (α = 1, 2, 3) fixing the location of an arbitrary point *M*(*y*_1_, *y*_2_, *y*_3_) in 3D space, the low indices determine the values of the power law exponents that correspond to the algebraic form of the fourth order. The choice of the sign’s combination (±) before the constants in (3) will be explained below. Three random sequences are determined by the values *r*_α*k*_ (α = 1, 2, 3; *k* = 1, 2, …, *N*). Expression (1) represents itself the complete form of the fourth order that contains the combination of three variables associated with an arbitrary point *M*(*y*_1_, *y*_2_, *y*_3_) and three arbitrary sequences *r*_α*k*_. The desired DGI is obtained from the following requirement:(4)$$\frac{1}{N}\sum_{k=1}^{N}L_{k}^{\left(4\right)}=I_{4}$$

In order to remove in Expression (4) the cubic terms, we introduce the variables
(5)$$Y_{\alpha }=y_{\alpha }-\langle r_{\alpha }\rangle ,\;\langle r_{\alpha }\rangle =\frac{1}{N}\sum_{k=1}^{N}r_{\alpha k}$$
and nullify the linear terms. This requirement helps us to separate the desired variables *Y*_α_ from each other and keep only the terms of the second and fourth orders, correspondingly. In order to decrease the number of constants in (3) and derive the DGI not depending on some additional constants, one defines three key ratio constants *R*^(α,β)^, with (α,β) = (1,2), (1,3), (2,3),
(6)$$\begin{array}{l} R^{\left(\alpha ,\beta \right)}=\frac{B^{\left(\alpha ,\beta \right)}}{A}=\frac{C^{\gamma \left(\alpha ,\beta \right)}}{A}=\frac{D^{\left(\alpha ,\beta \right)}}{A},\\ A_{40}^{\left(\alpha \right)}=A_{40}^{\left(\beta \right)}=A_{40}^{\left(\gamma \right)}\equiv A,\;\alpha ,\beta ,\gamma =1,2,3. \end{array}$$

It is convenient also to introduce the following notations for the integer moments and their intercorrelations and present them as
(7)$$\begin{array}{l} Q_{\alpha^{n}\beta^{m}\gamma^{l}}=\frac{1}{N}\sum_{k=1}^{N}\left({\left(\Delta r_{3k}\right)}^{m}{\left(\Delta r_{2k}\right)}^{n}{\left(\Delta r_{1k}\right)}^{l}\right)\equiv \langle {\left(\Delta r_{\alpha }\right)}^{m}{\left(\Delta r_{\beta }\right)}^{n}{\left(\Delta r_{\gamma }\right)}^{l}\rangle ,\\ \alpha \geq \beta \geq \gamma ,\;\left(\alpha ,\beta ,\gamma \right)=1,2,3. \end{array}$$

In the result of the introduced notations (6) and (7), the system of linear equations for the finding of unknown ratios *R*^(α,β)^ from the nullification requirement of the entering linear terms accepts the following form:(8)$$\begin{array}{l} \left[2Q_{221}-Q_{332}+\frac{3}{2}Q_{211}+\frac{1}{2}Q_{222}\right]\cdot R^{\left(1,2\right)}+\\ +\left[2Q_{331}-Q_{322}+\frac{3}{2}Q_{311}+\frac{1}{2}Q_{333}\right]\cdot R^{\left(1,3\right)}-2Q_{321}\cdot R^{\left(2,3\right)}=4Q_{111},\\ \left[2Q_{211}-Q_{331}+\frac{3}{2}Q_{221}+\frac{1}{2}Q_{111}\right]\cdot R^{\left(1,2\right)}-2Q_{321}\cdot R^{\left(1,3\right)}+\\ +\left[2Q_{332}-Q_{311}+\frac{3}{2}Q_{322}+\frac{1}{2}Q_{333}\right]\cdot R^{\left(2,3\right)}=4Q_{222},\\ -2Q_{321}\cdot R^{\left(1,2\right)}+\left[2Q_{311}-Q_{221}+\frac{3}{2}Q_{331}+\frac{1}{2}Q_{111}\right]\cdot R^{\left(1,3\right)}+\\ +\left[2Q_{322}-Q_{211}+\frac{3}{2}Q_{332}+\frac{1}{2}Q_{222}\right]\cdot R^{\left(2,3\right)}=4Q_{333}. \end{array}$$

The linear system of equations helps to reduce three moments (*Q*_333_, *Q*_222_, *Q*_111_) and seven intercorrelations of the third order (*Q*_332_, *Q*_322_, *Q*_221_, *Q*_211_, *Q*_331_, *Q*_311_, *Q*_321_) to the calculation of three unknown ratios *R*^(α,β)^ only. We should notice also that the combination of the algebraic signs in (3) is chosen in that way for the keeping of the partial solution *R* = 1 of system (8) in the case when all three random sequences *r*_α*k*_ are identical to each other, i.e., *r*_1*k*_ = *r*_2*k*_ = *r*_3*k*_. It is natural to define it as the case of spherical symmetry. If only two sequences coincide with each other (for example, *r*_1*k*_ = *r*_2*k*_ ≠ *r*_3*k*_), then we deal with the case of the cylindrical symmetry. In this case, the linear system (8) is reduced to a couple of linear equations relatively the variables *R*^(1,2)^ ≠ *R*^(1,3)^ = *R*^(2,3)^. After averaging procedure applied to Expression (4), the structure of the fourth-order form can be rewritten as
(9)$$K_{4}\left(Y_{1},Y_{2},Y_{3}\right)+K_{2}\left(Y_{1},Y_{2},Y_{3}\right)=I_{4}$$

The fourth- and the second-order forms entering to the left-hand side can be presented as
(10a)$$\begin{array}{l} K_{4}\left(Y_{1},Y_{2},Y_{3}\right)=Y_{1}^{4}+Y_{2}^{4}+Y_{3}^{4}+R^{\left(1,2\right)}Y_{1}Y_{2}\left[Y_{3}^{2}-\frac{1}{2}{\left(Y_{1}+Y_{2}\right)}^{2}\right]+\\ +R^{\left(1,3\right)}Y_{1}Y_{3}\left[Y_{2}^{2}-\frac{1}{2}{\left(Y_{1}+Y_{3}\right)}^{2}\right]+R^{\left(2,3\right)}Y_{2}Y_{3}\left[Y_{1}^{2}-\frac{1}{2}{\left(Y_{2}+Y_{3}\right)}^{2}\right]. \end{array}$$
(10b)$$\begin{array}{l} K_{2}\left(Y_{1},Y_{2},Y_{3}\right)=A_{11}Y_{1}^{2}+A_{22}Y_{2}^{2}+A_{33}Y_{3}^{2}+\\ \;+A_{12}Y_{1}Y_{2}+A_{13}Y_{1}Y_{3}+A_{23}Y_{2}Y_{3}. \end{array}$$

The constants *A*_αβ_ figuring in Expression (10b) are defined as
(11)$$\begin{array}{l} A_{11}=6Q_{11}-\left(Q_{22}+\frac{3}{2}Q_{21}\right)R^{\left(1,2\right)}-\left(Q_{33}+\frac{3}{2}Q_{31}\right)R^{\left(1,3\right)}+Q_{32}R^{\left(2,3\right)},\\ A_{22}=6Q_{22}-\left(Q_{11}+\frac{3}{2}Q_{21}\right)R^{\left(1,2\right)}+Q_{31}R^{\left(1,3\right)}-\left(Q_{33}+\frac{3}{2}Q_{32}\right)R^{\left(2,3\right)},\\ A_{33}=6Q_{33}+Q_{21}R^{\left(1,2\right)}-\left(Q_{11}+\frac{3}{2}Q_{31}\right)R^{\left(1,3\right)}-\left(Q_{22}+\frac{3}{2}Q_{32}\right)R^{\left(2,3\right)},\\ A_{12}=-\left(4Q_{21}+\frac{3}{2}Q_{11}+\frac{3}{2}Q_{22}-Q_{33}\right)R^{\left(1,2\right)}+2Q_{32}R^{\left(1,3\right)}+2Q_{31}R^{\left(2,3\right)},\\ A_{13}=2Q_{32}R^{\left(1,2\right)}-\left(4Q_{31}+\frac{3}{2}Q_{11}+\frac{3}{2}Q_{33}-Q_{22}\right)R^{\left(1,3\right)}+2Q_{12}R^{\left(2,3\right)},\\ A_{23}=2Q_{31}R^{\left(1,2\right)}+2Q_{21}R^{\left(1,3\right)}-\left(4Q_{32}+\frac{3}{2}Q_{22}+\frac{3}{2}Q_{33}-Q_{11}\right)R^{\left(2,3\right)}. \end{array}$$

The constant *I*_4_ (defined by 3 moments and 12 intercorrelations of the fourth order) figuring in the right-hand side of (9) is defined as
(12)$$\begin{array}{l} I_{4}=Q_{1111}+Q_{2222}+Q_{3333}-\left(Q_{2211}-Q_{3321}+\frac{1}{2}Q_{2111}+\frac{1}{2}Q_{2221}\right)R^{\left(1,2\right)}-\\ -\left(Q_{3311}-Q_{3221}+\frac{1}{2}Q_{3111}+\frac{1}{2}Q_{3331}\right)R^{\left(1,3\right)}-\\ -\left(Q_{3322}-Q_{3211}+\frac{1}{2}Q_{3222}+\frac{1}{2}Q_{3332}\right)R^{\left(2,3\right)}. \end{array}$$

It is interesting to notice that in the case of the spherical symmetry (*r*_1*k*_ = *r*_2*k*_ = *r*_3*k*_), all correlations coincide with each other and the value of *I*_4_ equals zero. The form of the fourth order (9) admits the separation of the variables in the spherical system of coordinates. If one accepts the following conventional notations: (13)$$\begin{array}{l} y_{1}=\langle y_{1}\rangle +R\sin\theta \cos\varphi ,\\ y_{2}=\langle y_{2}\rangle +R\sin\theta \sin\varphi ,\\ y_{3}=\langle y_{3}\rangle +R\cos\theta ,\;\\ 0\leq \theta \leq \pi ,\;0\leq \varphi \leq 2\pi ,\; \end{array}$$
then substitution of these variables into (9) leads to the following biquadratic equation relatively the unknown radius *R*(θ,φ):(14a)$${\left[R\left(\theta ,\varphi \right)\right]}^{4}+\left(\frac{P_{2}\left(\theta ,\varphi \right)}{P_{4}\left(\theta ,\varphi \right)}\right){\left[R\left(\theta ,\varphi \right)\right]}^{2}-\frac{I_{4}}{P_{4}\left(\theta ,\varphi \right)}=0.$$

The desired solution (*R*(θ, φ) > 0) is written as
(14b)$$R\left(\theta ,\varphi \right)={\left[\frac{\sqrt{P_{2}^{2}\left(\theta ,\varphi \right)+4I_{4}\cdot P_{4}\left(\theta ,\varphi \right)}-P_{2}\left(\theta ,\varphi \right)}{2P_{4}\left(\theta ,\varphi \right)}\right]}^{\frac{1}{2}}$$

The polynomials *P*_2,4_(θ,φ) entering in (14) are defined by the following expressions:(15a)$$\begin{array}{l} P_{4}\left(\theta ,\varphi \right)=\sin^{4}\theta \cdot \cos^{4}\varphi +\sin^{4}\theta \cdot \sin^{4}\varphi +\cos^{4}\theta +\\ +R^{\left(1,2\right)}\sin^{2}\theta \sin\varphi \cos\varphi \left[\cos^{2}\theta -\frac{\sin^{2}\theta }{2}{\left(\sin\varphi +\cos\varphi \right)}^{2}\right]+\\ +R^{\left(1,3\right)}\sin\theta \cos\theta \cos\varphi \left[\sin^{2}\theta \sin^{2}\varphi -\frac{1}{2}{\left(\sin\theta \cos\varphi +\cos\theta \right)}^{2}\right]+\\ +R^{\left(2,3\right)}\sin\theta \cos\theta \sin\varphi \left[\sin^{2}\theta \cos^{2}\varphi -\frac{1}{2}{\left(\sin\theta \sin\varphi +\cos\theta \right)}^{2}\right] \end{array}$$
(15b)$$\begin{array}{l} P_{2}\left(\theta ,\varphi \right)=A_{11}\sin^{2}\left(\theta \right)\cos^{2}\left(\varphi \right)+A_{22}\sin^{2}\left(\theta \right)\sin^{2}\left(\varphi \right)+\\ +A_{33}\cos^{2}\left(\theta \right)+A_{12}\sin^{2}\left(\theta \right)\sin\left(\varphi \right)\cos\left(\varphi \right)+\\ +A_{13}\sin\left(\theta \right)\cos\left(\theta \right)\cos\left(\varphi \right)+A_{23}\sin\left(\theta \right)\cos\left(\theta \right)\sin\left(\varphi \right). \end{array}$$

The last Expressions (13)–(15) determine the final form of the DGI in 3D space. It includes three surfaces determined by Expression (13). The further analysis shows that Expression (14b) equals zero (because *I*_4_ = 0) in the case of the coincidence of three compared random sequences (*r*_1*k*_ = *r*_2*k*_ = *r*_3*k*_). The radius *R*(θ,φ) can contain the complex expression when the integrand in (14b) becomes negative. It accepts the negative values when the constant *I*_4_ (which in most cases is defined by Expression (12)) becomes negative. In this case, it is convenient to rewrite expression (13) in the form
(16)y1=〈y1〉+|R(θ,φ)|sinθcosφ,y2=〈y2〉+|R(θ,φ)|sinθsinφ,y3=〈y3〉+|R(θ,φ)|cosθ,  |R(θ,φ)|=[Re(R(θ,φ))]2+[Im(R(θ,φ))]2,0≤θ<π,  0≤φ<2π. 

The curves defined by Equation (16) facilitate considerably the numerical analysis of initial data. As it follows from this preliminary analysis, this 3D surface is determined by the combination of 13 parameters: three moments of the first order <*y*_α_>, α = 1, 2, 3 from (3), six correlators of the second-order *A*_αβ_ from (11), three reduced correlators *R*^(α,β)^ from (8), and invariant of the fourth-order *I*_4_ from (12). We want to stress here again that the final Expressions (14) and (15) do not use any model and are determined completely by the measured data together with their measurement errors. Finishing this section, one can say that this method can be applied for the reduction of initial data. It is necessary to note that the dimension of the radius [*R*(θ,φ)] coincides with the dimension of initial data *y*_α_.

This reduction procedure can be divided into the following stages:

1. Initially, any available data can be written in the form of rectangle matrix [*N* × *M*], where number *N* (*j* = 1, 2, …, *N*—number of rows) determines the given data points and *M* (*m* = 1, 2, …, *M*—columns) determines the number of the repeated measurements forming in total the statistically significant sampling. As the result of the application of the 3D-DGI method, we obtain the reduced matrix [*M* × *S*], where each column of the reduced matrix (*Pr_m,s_*: <*y*_α_>(3), *R*^(α,β)^(3), A_αβ_(6), *I*_4_(1); α,β = 1, 2, 3) determines the complete combination of the moments and their intercorrelations (3 + 3 + 6 + 1 = 13) up to the fourth-order inclusive. In the result of the application of the 3D-DGI method, we obtain *s* = 1, 2, …, *S* (*S* = 13) distributions that demonstrate the variations of each statistical parameter *Pr_s_*(*m*) with respect to the number of repeated measurements (*m* = 1, 2 …, *M*).

2. The further reduction is possible if one takes into account that each random function *y_s_*(*m*) ≡ *Pr_s_*(*m*) is located inside the rectangle *M* × (Range[*y_s_*(*m*)]), where *Range*(*f*) = max(*f*) − min(*f*). For comparison of one random function *y*_1,*s*_(*m*) with another *y*_2,s_(*m*) corresponding to the chosen parameter *s* (*s* = 1, 2, …, *S*), one can use the following simple formula:
(17)$$Q_{1,2}(s)=\frac{Range(y_{1,s})+Range(y_{2,s})}{\max(y_{1,s},y_{2,s})-\min(y_{1,s},y_{2,s})}$$

This expression, in spite of its simplicity, is really effective for the comparison of the statistical closeness of a pair of random functions belonging to the given/another sampling participating in the comparison operation. Really, if the function *Q*_1,2_(*s*) is located in the interval [1,2], then the pair random functions are statistically close to each other. In the case when Q_1,2_(s) ∈ [0,1), one can conclude that the pair random functions compared are statistically different. Besides this important parameter (16), one can take into account the symmetry of the random function *y*_1_(*m*). Any random function located in the rectangle *M* × *Range*[*y*(*m*)] crosses the line <*Pr*(*m*)>, coinciding with its mean value. Therefore, for evaluation of the symmetry of a random function, one can introduce the value
(18)$$Sm(y)=\frac{\mathrm{mean}(y)-0.5\cdot \left(\max(y)+\min(y)\right)}{Range(y)}$$

If the value *Sm*(*y*) is located near zero (*Sm*(*y*) ≈ 0), then the line <*y*> divides the rectangle *M* × *Range*[*y*(*m*)] into two almost equal parts. In other cases, the value *Sm*(*y*) ∈ [−0.5, 0.5] determines the measure of asymmetry. After the application of Expression (17) for comparing similar columns (belonging to the same parameter *Pr_s_*), one can receive finally the vector of the length *S* = 13 that contains information about the statistical closeness of two matrices compared. It is interesting to notice that simple Expression (17) can be used also for comparison each successive measurement with another one in the given rectangle matrix [*N* × *M*]. If one compares the vectors *y_m_* forming the columns of the initial matrix with each other, then in the result of application (17), one can obtain the symmetrical matrix *U*(*m*_1_,*m*_2_) (*m*_1,2_ = 1, 2, …, *M*) with elements located in the interval 0 ≤ *U*(*m*_1_,*m*_2_) ≤ 2. Only elements located in interval 1 ≤ *U*(*m*_1_,*m*_2_) ≤ 2 will correspond to a “good” experiment, while the elements from interval 0 ≤ *U*(*m*_1_,*m*_2_) < 1 should be considered as possible “outliers” and correspond to “bad” experiment. 

3. How to find the parameters belonging only to one matrix in order to compare them with similar parameters of another tested matrix in cases when the reference matrix is absent? Initially, it is necessary to scale each column to the same interval
(19)$$\begin{array}{l} Prn_{s}=\frac{DPr_{s}}{Range({\mathrm{Pr}}_{s})}\equiv \frac{Pr_{s}-\langle Pr_{s}\rangle }{Range({\mathrm{Pr}}_{s})},\\ -\frac{1}{2}\leq Prn_{s}\leq \frac{1}{2},\;\langle Pr_{s}\rangle =\frac{1}{M}\sum_{m=1}^{M}Pr_{m,s}. \end{array}$$

This normalization procedure makes all parameters *Pr_s_* statistically close to each other with mean value equaled zero mean(*Prn_s_*) = 0 and with the *Range*(*Prn_s_*) = 1. If one integrates Expression (19) for each parameter one can receive the statistically different curves *JP_s_* = Integral(*Prn_s_*) for each initial parameter (*s* = 1, 2, …, *S*). The distributions of the ranges of these integral curves *P*_1_ = *Range*(*JP_s_*), together with the distribution of asymmetries *P*_2_ = *Range*(*Sm*(*JP_s_*)) calculated with the help of Expression (18), give finally the matrix containing [*S* × 2] containing (*S* = 13) rows and two columns only. If we calculate the ranges of these two columns, we obtain finally two values only that can characterize the initial matrix [*N* × *M*]. If we have a set of matrices [*N* × *M*]*_q_* (*q* = 1, 2, …, *Q*), then this simple and general procedure allows us to select the “best” one having minimal values of these two key parameters *P*_1,2_. It will characterize the stability of the initial sequence and their minimal values will serve as a criterion for the selection of the “best” TLS among other TLS(s). We want to emphasize here that this final stage of treatment of “big” matrices differs from the procedure used in the paper [29]. Earlier, one of us (RRN) had the set of rectangle matrices that can be characterized as “normal/reference” ones and compared them with “strange/tested” matrices associated with defects. Data that will be analyzed below do not contain this information. Therefore, we propose another procedure described above for the selection of the “best” data expressed in the form of rectangle matrices.

Concluding this section, it is necessary to mention the following. The POLS is the universal tool that can be applicable for smoothing all available data. It possesses some remarkable features that make it really universal for many applications, which include the following:The POLS is a linear tool, and it does not distort initial data;When the value of the smoothing window tends to zero (*w* → 0), then the smoothed replica coincides with the initial data. In another limiting case, when w >> 1, the smoothed replica coincides with its arithmetic mean.

Another new instrument as the 3D geometrical invariants can be defined as the “universal” tool as well, by virtue of the following features:Thanks to 13 universal parameters defining the feature space, it allows the comparison of the different random sequences having different natures;It can be applied to analyses of the TLS and, therefore, this tool forms a universal platform that cannot contain treatment errors;Section 3, given above, gives an example for its application to real data.

We should also note that both methods keep the units of the initial data.

## 4. Experimental Results and Their Treatment 

### 4.1. Experimental Setup 

In order to imitate the process of helicopter blade deformation measurement, an experimental setup was designed that included a composite plate with six strain detectors based on fiber Bragg gratings attached to the upper surface of the plate using epoxy adhesive. The FBGs used in the experiment were manufactured at the Department of Radiophotonics and Microwave Technologies of KNRTU-KAI (Kazan) using a common fabrication technique based on the Lloyd interferometer. Figure 3a presents a photograph of the experimental plate, and Figure 3b shows a schematic representation of the detector arrangement on the plate (positions 1.1–1.3, 2.1–2.3), as well as the main dimensions. During the experimental procedure, the plate was fixed on one end to the vibrating stand, while the opposite end remained free, mimicking the vibrations of a helicopter blade. To the free end of the plate, a load with predefined mass was attached at point *A* (see Figure 3b). The values of the Bragg wavelength shifts of all FBG detectors were acquired for seven different load cases (*L* = 0, 50, 60, …, 100 g). It must be noted that in this preliminary research, the authors did not pursue the goal to compare the vibrations with the real-life reference, which will be the aim of the subsequent studies. The measurements were taken during a certain time interval (~8 s) so that 800 points of data were registered with the rate of ~100 Hz for each load case. The data were collected using an FBG interrogation device developed at KNRTU-KAI based on the Ibsen I-MON 512 interrogation monitor. 

### 4.2. Treatment Procedure 

Based on the treatment procedure explained in Section 3, we received seven matrices. Each matrix included 6 columns and 800 data points corresponding to different loads (*L* = 0, 50, 60, …, 100 g), which were subjected to the same vibrations measured in nm as the wavelength shifts of the fiber-optic detectors. Each column inside the matrix 800 × 6 included the recorded vibrations associated with the fixed detector. Each detector shown in Figure 3b was tightly associated with a number of columns (their correspondence to each other shown below), and it was located on the plate imitating the helicopter blade. 

In order to select the desired value of the smoothing window *w* figuring in Expression (2), we chose a simple criterion that the low-frequency trends obtained after POLS should be almost uncorrelated with each other. If we fix the interval of correlations [−0.5 × 10^−3^, 0.5 × 10^−3^] for the smoothed trends and require that
(20)$$-0.0005\leq \frac{Ysm(0,w)\cdot Ysm(L,w)}{\sqrt{{\left(Ysm\left(0,w\right)\right)}^{2}{\left(Ysm\left(L,w\right)\right)}^{2}}}\leq 0.0005,$$
where *Ysm*(*L*,*w*) is determined by Expression (2) and corresponds to the smoothed function for the given load *L* = 0, 50, 60, …, 100 g, then from condition (20), one can find the desired value *w* from the interval [100–200]. We chose the value *w* = 150, corresponding to the middle of the found interval. The figures below demonstrate the low-frequency fluctuations that were obtained for detectors D-1.1 and D-2.2 for comparison with Figure 4a,b. We notice that after application of the POLS the smoothed curve associated with fluctuations with load is shifted up. It takes place because condition (20) allows decreasing the value of the initial correlations.

Figure 5a demonstrates the results of the application POLS for *w* = 150 and detector D-1.1. Figure 5b shows the same result for D-2.2 at *w* = 150, as well. Analysis of these curves shows that for construction of the calibration curves showing the dependence of each detector with respect to the applied load one can select the differences between mean values corresponding to these smoothed curves.

Figure 6a demonstrates six calibration curves calculated for all types of detectors. All curves are monotone and we see the increase of these curves with respect to the applied load. Figure 6b comprises the data calculated with respect to Expression (17). It demonstrates the sensitivity of different detectors with respect to the applied load. Three detectors D-2.3, D-1.1, and D-2.2 are the most sensitive, while the detector D-2.1 has minimal sensitivity. The detectors D-1.2 and D-1.3 have an intermediate sensitivity. We highlight again that the mean values conserve the vibration units.

It is interesting to analyze also 13 parameters (given by Expressions (8), (11), and (12)) that form the desired 3D surface. In order to compare these parameters with each other, we form the corresponding SRAs after applying the POLS procedure for different loads. Figure 7a demonstrates 3 curves calculated for the load L = 0 g. In order to notice the possible monotone behavior of these curves, we use the corresponding SRAs for different loads. On the right-hand side (Figure 7b), we place similar triple curves corresponding to L = 50 g. Other curves for the loads L = 60–100 g look similar and therefore are not shown.

In order to compare these parameters with each other, we take the mean values for the correlators of the first order (corresponding to the gravity center), then correlators of the second, third, and fourth orders, correspondingly. For us, it is important to detect a monotone behavior among these correlators. Figure 8a demonstrates six calibration curves calculated for the mean values of the first-order correlators. All curves are almost monotone, and we see an increase in these curves with respect to the applied load.

However, the detectors D-2.1 D-1.2 and D-1.3 show nonmonotonic behavior in the limits of the load 0–50 g. We associate this phenomenon with the influence of the vibration waves amplitudes that are propagated over the plate. This phenomenon needs more detailed research. These curves are similar to Figure 6a. Figure 8b shows the nonmonotonic behavior of the correlators of the second order. In comparison with Figure 8a, the behavior of the correlators of the second order is different. As one can see, for this figure, the values of these correlators are small; only detector D-2.2 demonstrates the high values of the correlators of the second order when the applied load is absent.

Figure 9a demonstrates six calibration curves calculated for mean values of the third-order correlators. These curves have different behavior if we compare them with Figure 8. On the small figure inside, we show the correlators of the third order for detectors D-2.2 and D-2.3 that have a monotonic behavior. Figure 9b shows the different behavior of the correlators of the fourth order, in comparison with Figure 9a. As one can see, for this figure, the values of these correlators are small. However, only detectors D-2.2, D-1.1, and D-2.1 demonstrate the relatively high values of the correlators of the fourth order when the applied load is absent.

Analysis of these figures shows that they have different sensitivity to the applied load. Only the mean correlators of the first order having monotonic property can be used for calibration purposes. It presents an interesting possibility to give also the plots of 3D surfaces at different loads. It will reflect a unique combination of the 13 correlation parameters and this surface will serve as a *specific fingerprint* facilitating an initial analysis of the available data. Figure 10a shows the 3D surface for the L = 0 g, while Figure 10b presents the surface for L = 50 g. The 3D surface for the L = 90 g is demonstrated in Figure 11a. Figure 10b shows the surface for L = 100 g. In all these 3D surfaces, the unit of the OZ axis is conserved, and it is given in the units coinciding with initial data (values of vibrations in our case). As one can notice from the comparison of these figures, they are different in their values and have different forms.

## 5. Discussion and Conclusions

The proposed data treatment method based on the procedure of the optimal linear smoothing (POLS) allowed us to eliminate the noise induced by vibrations of the experimental plate, imitating the helicopter blade in flight conditions. As the result of the treatment procedure, monotonic calibration curves were obtained for each detector, making it possible to distinguish the increase of load applied to the plate with an increment of 10 g, which otherwise would be impracticable. It should be noted that the calibration curves comprising mean values of POLS-treated wavelength shift data of the detectors (Figure 6a) demonstrated higher mean deformation of the detector 2.2 (and even higher in case of the detector 2.3) than the one of the detector 2.1, which does not correspond to the a priori assumption that the detectors 2.1 and 1.1 (which are the nearest to the fixed end of the plate) should be subjected to the highest deformations, while 2.3 and 1.3 should experience the lowest strain. More detailed analysis based on a complete set of data allows concluding that detector 2.1 had lower sensitivity, in comparison with others, which can be caused by relatively weak adhesive bonding to the surface of the plate. 

Highly likely, this peculiarity is related to the chosen material and formation in it the deformation waves. 

However, the calibration curves calculated for mean values of the first-order correlators (Figure 8a) do agree with the mentioned assumption, showing the higher values at the detectors 1.1 and 2.1, and the lowest ones at the detectors 1.3 and 2.3.

As one can infer from the content given above, the second method associated with 3D invariants takes into account 13 quantitative parameters associated with a combination of different moments and their intercorrelations up to the fourth-order inclusive. They reflect more sophisticated behavior of the moments and their intercorrelations evoked by the applied load. Figure 8b and Figure 9a,b show their peculiarities. These 13 parameters allow the calculation of the desired 3D surface that can serve as a specific fingerprint, differentiating one set of initial data from another one. Figure 10a,b and Figure 11a,b vividly demonstrate these peculiarities. 

Therefore, the combination of these two methods that were used to analysis of real load data can be applied successfully to other data, because they are general, do not use a specific model, and are free from treatment errors. 

## Figures and Tables

**Figure 1 sensors-21-04028-f001:**
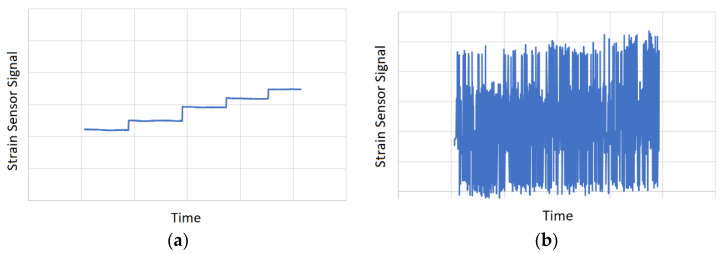
Example of data obtained by a strain sensor located on a blade for five cases of vertical loads: (**a**) without oscillations; (**b**) with oscillations.

**Figure 2 sensors-21-04028-f002:**
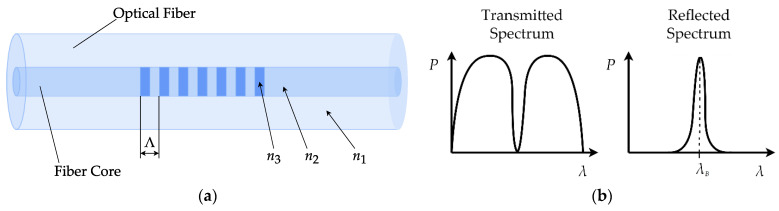
(**a**) Schematic representation of a fiber Bragg grating: Λ is the grating period, *n*_1_ is the refractive index of the fiber cladding, *n*_2_ is the refractive index of the fiber core, and *n*_3_ is the refractive index of periodic variations forming the FBG; (**b**) spectra of transmitted and reflected light of an FBG.

**Figure 3 sensors-21-04028-f003:**
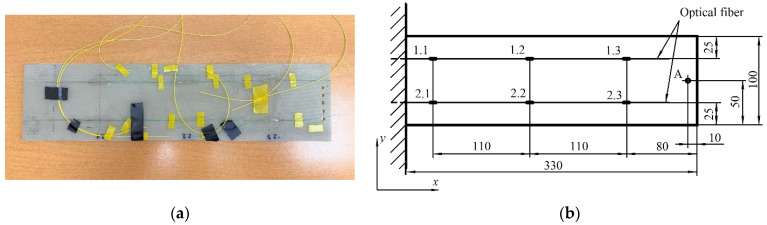
Experimental plate with six FBG detectors: (**a**) photograph; (**b**) schematic representation of the plate with the positions of the detectors (1.1–1.3, 2.1–2.3), and the position of the load application (point A) (dimensions are in mm).

**Figure 4 sensors-21-04028-f004:**
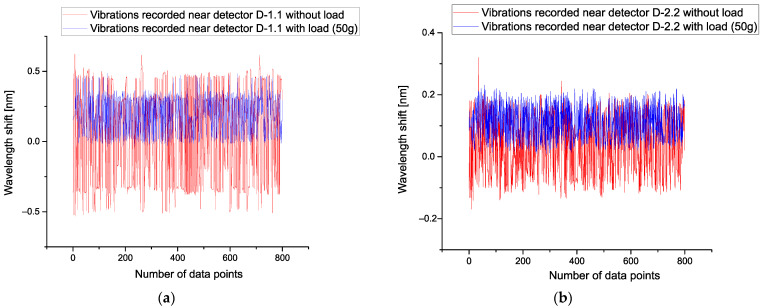
(**a**) This figure shows the level of vibrations without loads (red lines) and with minimal load corresponding to the load equal to 10 g (blue lines). This figure reflects the noise fluctuation for the first detector D-1.1. In order to stress their difference, the same picture Figure 4 (**b**) is placed on the right-hand side for detector D-2.2 (with load 10 g) and without load (red lines), correspondingly. For other detectors, the obtained pictures look the same and therefore they are omitted.

**Figure 5 sensors-21-04028-f005:**
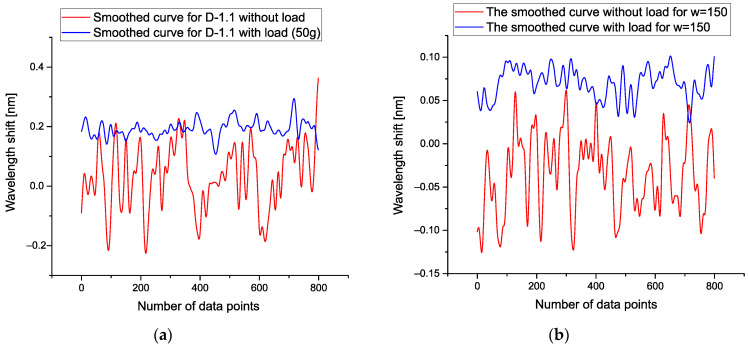
(**a**) Results of the application of POLS for *w* = 150 and detector D-1.1; (**b**) the same results for D-2.2 at *w* = 150.

**Figure 6 sensors-21-04028-f006:**
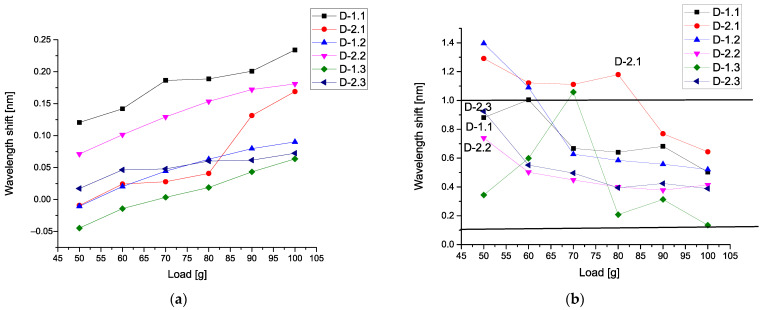
(**a**) Six calibration curves (mean values) calculated for all types of detectors; (**b**) comparison of correlators calculated for different detectors.

**Figure 7 sensors-21-04028-f007:**
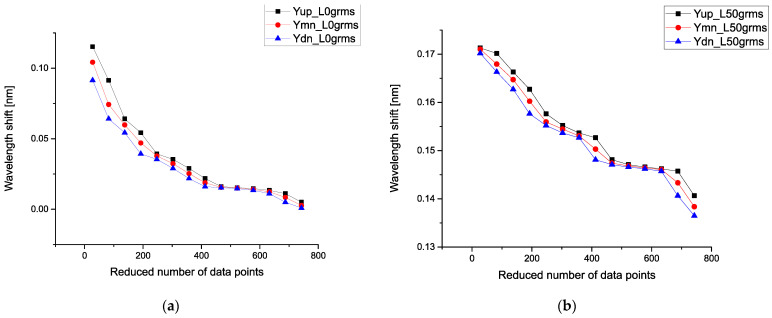
(**a**) Sequences of the ranged amplitudes (SRAs) calculated for the load L = 0 g; (**b**) curves of SRAs corresponding to L = 50 g. Other curves for the loads L = 60–100 g look similar and therefore are not shown. Black markers (Yup) denote the distribution of maximum values, red markers (Ymn) represent the distribution of mean values, and blue markers (Ydn) show the distribution of minimum values.

**Figure 8 sensors-21-04028-f008:**
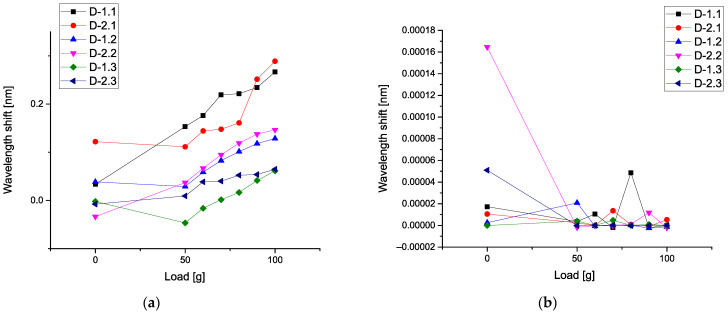
(**a**) Calibration curves calculated for mean values of the first-order correlators; (**b**) mean values of the second-order correlators.

**Figure 9 sensors-21-04028-f009:**
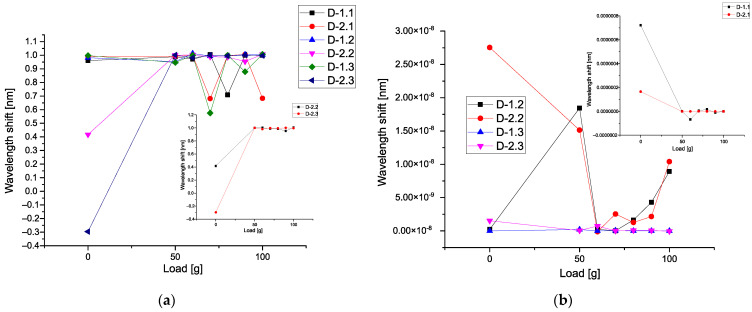
(**a**) Calibration curves calculated for mean values of the third-order correlators; (**b**) calibration curves calculated for mean values of the fourth-order correlators.

**Figure 10 sensors-21-04028-f010:**
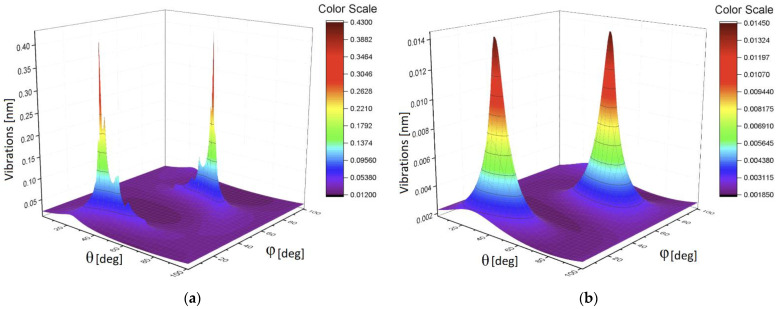
(**a**) The 3D surface for the L = 0 g; (**b**) the 3D surface for L = 50 g.

**Figure 11 sensors-21-04028-f011:**
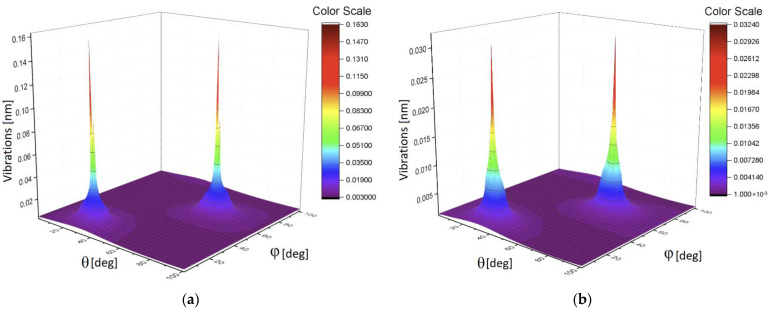
(**a**) The 3D surface for the L = 90 g; (**b**) the 3D surface for L = 100 g.

## Data Availability

The data presented in this study are available on request from the corresponding author. The data are not publicly available due to the rules of our contract conditions with our customer.

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
