# Peer review of "Noise Cancellation of Helicopter Blade Deformations Measurement by Fiber Bragg Gratings"

_sensors, 2021, doi:10.3390/s21124028_

Round 1
Reviewer 1 Report
The paper shows a theoretical and experimental approach that could be applied to eliminate the noise of the deformations of a helicopter blade.
The paper is focused on two main areas: mathematical models and use of the simulations to analyze the experimental data.
The modelling work is based on the use of the procedure of the optimal linear smoothing (POLS) and a complete Discrete Geometrical Invariant. However, the final conclusion of the virtues of this numerical approach is not clear and need to be more specifically justified (at the end of Section 3.2, for example).
The experimental work is based on the data recorded by Fiber Bragg gratings when submitted to loads. The experimental set-up aims to simulate the loads and vibrations suffered by one helicopter blade; however, the composite plate used looks a very simplified model to be able to be compared with a helicopter blade and no reference to the real vibrations encountered during the flight are referenced. In my opinion, this could be improved and clarified in Section 4.1 “Experimental setup”.
The references of Figures 5 to 11 within the text are missing. This fact complicates the reading and analysis of the paper. The text should be revised from lines 361 to lines 408 to address this question and keep the format of a scientific paper in this sense.
Moreover, the units of axes X and Y in all the figures need to be reviewed; the data should include physical quantities in the International System of Units (SI) or arbitrary units (a.u.) and not “behavior”, “comparison”, “total set”…
As far as the paper is concerned, I would first recommend a careful formatting of the text in “Section 4.2. Treatment procedure” and the data contained in the figures before publishing.
Specific comments:
In my opinion, the paper should also address the following suggestions:
- The Introduction Section is well organized and driven. However, the insertion of Figures in the Introduction Section is not recommended. Figure 1 should be removed and paragraph from line 100 to line 116 should be revised.
- Line 125: Consider to change the phrase to “inscribing A systematic variation…” instead of using “the”.
- Line 130-132: the Bragg wavelength of a FBG is shifted when affected to more parameters than strain and temperature. Please, include more parameters than can influence the response for a FBG and a reference with a review paper with the latest works and advances; the work in DOI:10.1117/1.OE.59.6.060901 is a suggestion.
- Line 138: their usage is 137 limited in some applications due to the complexity and high cost of interrogators”. It is not clear with this phrase if the papers referenced below in the text use interrogators or not. Consider to eliminate this phrase.
- After the revision of the different types and methods to use FBG for deformation measurements, a last paragraph after line 156 referring to the method used in this work would be useful to help the reading of the paper.
Consider to write a paragraph before Section 3 that includes the type of FBG and method of interrogation used in this paper.
- Lines 159-161: references 22-27 are related with the use of optimal linear smoothing in different applications and areas of interest. A brief description of the referenced papers and the advantage to use this numerical technique is recommended.
- Line 183. Reference [28] is missing.
- Lines 189-191: mistake in expression number? I think that the expression/equation mentioned here should be number 3, not number 1.
- Lines 200-202: Long sentence, make it clearer by inserting “,” in the text, for example.
- Lines 248-249: review sentence, it is not clear.
- Line 333: specify the “time interval” and the rate of recorded data with the optical interrogator. This can be an important number depending on the value of the vibrations frequencies.
- Line 342: related to the above comment. It is indicated that all the loads are “subjected to the same vibrations”. Clarify the level of the vibrations suffered by the plate for all the loads cases.
- Figure 4. Include a figure with the data of the wavelength shift measured with the interrogator unit. These data helps to estimate the limit of resolution in terms of optical wavelength data.
- Lines 397-402: it is mentioned that only the “mean correlators of the first order” can be used for calibration purposes. It is not clear how this limitation affects the use of this technique in the overall result. Please, clarify.
Author Response
First of all, we wish to thank the anonymous Reviewer for the precious comments that allowed us to improve the quality of the paper.
The modelling work is based on the use of the procedure of the optimal linear smoothing (POLS) and a complete Discrete Geometrical Invariant. However, the final conclusion of the virtues of this numerical approach is not clear and need to be more specifically justified (at the end of Section 3.2, for example).
Thank you for the valuable comment. The following text was added at the end of Section 3.2 (lines 328–342): “Concluding this section, it is necessary to say the following. The POLS is the universal tool that can be applicable for smoothing of all available data. It possesses some remarkable features that make it really universal for many applications:
1. The POLS is linear tool and it does not distort initial data.
2. When the value of the smoothing window tends to zero (w → 0) then the smoothed replica coincides with the initial data. In another limiting case, when w >>1, the smoothed replica coincides with its arithmetic mean.
Another new instrument as the 3D-geometrical invariants can be defined as the “universal” tool as well, by virtue of the following features:
1. Thanks to 13 universal parameters defining the feature space it allows to compare the different random sequences having different nature.
2. It can be applied to analysis of the TLS and, therefore, this tool forms a universal platform that cannot contain the treatment errors.
3. The section 3, given above, gives an example for its application to real data.
We should notice also that the both methods keep the units of the initial data.”
The experimental work is based on the data recorded by Fiber Bragg gratings when submitted to loads. The experimental set-up aims to simulate the loads and vibrations suffered by one helicopter blade; however, the composite plate used looks a very simplified model to be able to be compared with a helicopter blade and no reference to the real vibrations encountered during the flight are referenced. In my opinion, this could be improved and clarified in Section 4.1 “Experimental setup”.
We thank the Reviewer for the suggestion. The aim of the experiment was to obtain a set of data with noise induced by vibrations similar to the one that can be obtained from a rotating helicopter blade during flight in order to demonstrate the principle of the proposed data treatment approach.
In this preliminary research, the authors did not pursue the goal to compare the vibrations with the real-life reference, which will be the aim of the subsequent studies. We clarified this in Section 4.1 (lines 357–360).
The references of Figures 5 to 11 within the text are missing. This fact complicates the reading and analysis of the paper. The text should be revised from lines 361 to lines 408 to address this question and keep the format of a scientific paper in this sense.
Thank you for the valuable remark. The text was revised accordingly (highlighted in yellow).
Moreover, the units of axes X and Y in all the figures need to be reviewed; the data should include physical quantities in the International System of Units (SI) or arbitrary units (a.u.) and not “behavior”, “comparison”, “total set”… As far as the paper is concerned, I would first recommend a careful formatting of the text in “Section 4.2. Treatment procedure” and the data contained in the figures before publishing.
Thank you for your concern. The units to Figures 5-11 are clear and, from our point of view, they do not need additional comments. However, the authors have added physical quantities to the figures where applicable. The vibrations are measured in nm as the wavelength shifts of fiber Bragg gratings used as detectors. As it has been mentioned above, both of the methods POLS and 3D-DGI method conserve the units of the initial data. It is easy to notice the conservation of the units over all the figures 5-11.
Specific comments:
In my opinion, the paper should also address the following suggestions:
- The Introduction Section is well organized and driven. However, the insertion of Figures in the Introduction Section is not recommended. Figure 1 should be removed and paragraph from line 100 to line 116 should be revised.
We thank Reviewer for his/her appreciation. As far as the authors are concerned, Figure 1 is very important for explanation of the aim of the current work, therefore it was not removed.
- Line 125: Consider to change the phrase to “inscribing A systematic variation…” instead of using “the”.
Thank you for the remark. The sentence was corrected.
- Line 130-132: the Bragg wavelength of a FBG is shifted when affected to more parameters than strain and temperature. Please, include more parameters than can influence the response for a FBG and a reference with a review paper with the latest works and advances; the work in DOI:10.1117/1.OE.59.6.060901 is a suggestion.
The working principle of FBG was clarified and the reference was added as follows: “The working principle of FBG is based on the change of either the effective refractive index (neff) or the grating period (Λ) and the resulting shift of its central wavelength (λB) when the sensor is subjected to physical fields, such as strain, temperature, pressure, humidity, etc. [14].” (lines 130 – 133).
- Line 138: their usage is 137 limited in some applications due to the complexity and high cost of interrogators”. It is not clear with this phrase if the papers referenced below in the text use interrogators or not. Consider to eliminate this phrase.
We thank the Reviewer for the remark. The phrase was eliminated, and the following sentence was added: “The common disadvantage of the mentioned interrogation techniques is the complexity and high cost of devices required for their implementation.” (lines 141 – 143)
- After the revision of the different types and methods to use FBG for deformation measurements, a last paragraph after line 156 referring to the method used in this work would be useful to help the reading of the paper. Consider to write a paragraph before Section 3 that includes the type of FBG and method of interrogation used in this paper.
Thank you for the suggestion. The following text was added before Section 3 (lines 157 – 161): “The FBG interrogation device used in the current work implemented conventional wavelength-division multiplexing of FBG-sensors, where each grating had different Bragg wavelength so that their spectral responses did not superimpose on each other during measurements. The device was based on the Ibsen® I-MON spectrum analyzer and was operated by a custom software.”
- Lines 159-161: references 22-27 are related with the use of optimal linear smoothing in different applications and areas of interest. A brief description of the referenced papers and the advantage to use this numerical technique is recommended.
We thank the Reviewer for the concern. The explanation of the POLS is simple. It is based on expression and associated with the selection of the optimal smoothing window. In our case, the criterion is based on the Pearson correlation coefficient. See expression (20), for example. The detailed description of the POLS procedure will occupy large space, and therefore we omit the details.
- Line 183. Reference [28] is missing.
Thank you for the remark. The reference was corrected.
- Lines 189-191: mistake in expression number? I think that the expression/equation mentioned here should be number 3, not number 1.
We thank the Reviewer for the observation. The expression number was corrected.
- Lines 200-202: Long sentence, make it clearer by inserting “,” in the text, for example.
Thank you for the suggestion. The comma was inserted in the text.
- Lines 248-249: review sentence, it is not clear.
The sentence was modified as follows: “It accepts the negative values when the constant I4 (which in most cases is defined by expression (12)) becomes negative.”
- Line 333: specify the “time interval” and the rate of recorded data with the optical interrogator. This can be an important number depending on the value of the vibrations frequencies.
Thank you for the suggestion. The sentence was supplemented as follows: “The measurements were taken during a certain time interval (~8 s), so that 800 points of data were registered with the rate of ~100 Hz for each load case.”
- Line 342: related to the above comment. It is indicated that all the loads are “subjected to the same vibrations”. Clarify the level of the vibrations suffered by the plate for all the loads cases.
Thank you for the concern. The vibrating stand was generating the same level of vibrations during the measurements for all the load cases. The free end of the plate was oscillating with the amplitude of ⁓30 mm without load applied to it, and the amplitude of the oscillations decreased to ⁓12 mm for the highest load case. However, the experiment was not aimed at the measurement of the vibration level, since it was designed solely for the acquisition of the “noisy” strain signal.
- Figure 4. Include a figure with the data of the wavelength shift measured with the interrogator unit. These data helps to estimate the limit of resolution in terms of optical wavelength data.
We thank the Reviewer for the concern. The data shown in Figure 4 are presented in terms of the FBG wavelength shift of the Detector 1-1. The unit of measure was added in the figure.
- Lines 397-402: it is mentioned that only the “mean correlators of the first order” can be used for calibration purposes. It is not clear how this limitation affects the use of this technique in the overall result. Please, clarify.
The additional research shows that the mean correlations of the first order are important. The invariant correlations of the higher orders are taken into account by the second method. The figures 8(b) and figures 9(a,b) show the difference between the correlators of the different orders.

Reviewer 2 Report
Comments:
This manuscript presents the optimal linear smoothing and 3D-invarinat methods for elimination of noise in the strain measurement data caused by the blade oscillations, thereby enhancing metrological performance of the helicopter weight measurement system. I think some descriptions are unclear and the discussions are inadequate. I have several suggestions:
- Can the authors give a brief introduction on the 3D-invariant's role in noise cancellation?
- Figure 1: Suggest adding the units of abscissa and ordinate;
- Line 183 and 316: Ref. 28 do not exist;
- Line 253: Suggest giving 13 parameters more clearly;
- Figure 5: In Line 364 and 365, the authors point out that w=150. But in Figure 5, w=15. Which one is right? Besides, please indicate that the load in Figure 5(b).
- Line 361: The authors point out that the low-frequency fluctuations were obtained. I suggest adding a comparison of the frequency spectrum before and after noise cancellation;
- Figure 6: D-2.3, D-1.1 and D-2.2 are the most sensitive. Please give more discussions;
- Figure 7: Suggest explaining the relationship between sequences of the ranged amplitudes (SRA) and the 13 parameters;
- Figure 7: What do Yup, Ymn and Ydn stand for?
- Figure 8(a): The authors point out that all curves are monotone. But in my point of view, D-2.1, D-1.2, and D-1.3 are not monotone. Please discuss this difference;
- Figure 10 and Figure 11: Please discuss the results of these two figures in detail and explain their relationship to noise cancellation.
- There is no introduction of the helicopter blade’s structure. Is it suitable to use the rectangular plate to do the experiment? More discussions are necessary.
Author Response
We would like to thank the Reviewer for the comprehensive review of the paper and the valuable remarks.
1. Can the authors give a brief introduction on the 3D-invariant's role in noise cancellation?
We thank the Reviewer for the concern. The aim of the noise cancellation in this work is to make it possible to distinguish between various loads, which otherwise would not be possible to do from the noisy signal. As the authors state in lines 481 – 484, the 3D-surfaces calculated from the 13 parameters can serve as a specific fingerprint differentiating one set of initial data from another one. The figures 10(a,b) and 11(a,b) vividly demonstrate these peculiarities. The surfaces change their form at various loads and indicate to researcher that these changes can be differentiated in terms of 13 parameters forming a feature space.
2. Figure 1: Suggest adding the units of abscissa and ordinate
Thank you for the remark. The purpose of Figure 1 is to present a qualitative characteristic of strain measurements that can be obtained by any type of strain sensor, such as an electrical resistance strain gauge or a fiber Bragg grating. So, the units can vary depending on the type of sensor used, and therefore are omitted in the figure.
3. Line 183 and 316: Ref. 28 do not exist;
We thank the Reviewer for the remark. The reference number was corrected.
4. Line 253: Suggest giving 13 parameters more clearly;
These 13 parameters include the moments and intercorrelations up to the fourth order inclusive. The sentence was restated as follows (lines 259-261): “…three moments of the first order <yα>, α=1,2,3 from (3), six correlators of the second order Aαβ from (11), three reduced correlators R(α,β) from (8), and invariant of the fourth order I4 from (12).”
5. Figure 5: In Line 364 and 365, the authors point out that w=150. But in Figure 5, w=15. Which one is right? Besides, please indicate that the load in Figure 5(b).
Thank you for the observation. The value was corrected in the figure (w=150).
6. Line 361: The authors point out that the low-frequency fluctuations were obtained. I suggest adding a comparison of the frequency spectrum before and after noise cancellation;
If we compare Figs. 4(a,b) with Figs 5(a,b), one can notice that the POLS removes the high-frequency fluctuations (the scale of abscissa is the same in both figures). We do not see the reason to add the corresponding figures proving that the high-frequency fluctuations are actually removed, because that does not contribute to the further research discussed in the paper.
7. Figure 6: D-2.3, D-1.1 and D-2.2 are the most sensitive. Please give more discussions;
As far as the authors are concerned, the propagation of vibration waves over the plate is not uniform, we have the places where these waves are concentrated (the antinodes), and discharging places. The location of detectors (see Fig.3(b)) is uniform and therefore they have different sensitivities to propagation of vibration waves.
8. Figure 7: Suggest explaining the relationship between sequences of the ranged amplitudes (SRA) and the 13 parameters;
The 3D-DGI method is really universal and can be applied to any three sequences with calculation of the desired 13 parameters. On the other hand, the SRA defines the ordered sequence that includes in itself all possible N! permutations of random realizations of the sequence having initially N data points. Therefore, from our point of view, the procedure of reduction to three incident points shown in Figs. 7(a,b) with subsequent calculation of three statistically closed sequences is optimal.
9. Figure 7: What do Yup, Ymn and Ydn stand for?
Thank you for the question. The following was added to the figure description (lines 415 – 417): “Black markers (Yup) denote the distribution of maximum values, red markers (Ymn) represent the distribution of mean values, blue markers (Ydn) show the distribution of minimum values.”
10. Figure 8(a): The authors point out that all curves are monotone. But in my point of view, D-2.1, D-1.2, and D-1.3 are not monotone. Please discuss this difference;
Possible explanation of this phenomenon is added (lines 426-428).
11. Figure 10 and Figure 11: Please discuss the results of these two figures in detail and explain their relationship to noise cancellation.
Lines 477-484 explain the reason why we show the surfaces in Figs.10(a,b)-11(a,b). Showing these surfaces, we want to demonstrate their sensitivity expressed in terms of 13 invariant parameters to different loads. They change their form and indicate to researcher that these changes can be differentiated in terms of 13 parameters forming a feature space. These differences are shown in the previous figures.
12. There is no introduction of the helicopter blade’s structure. Is it suitable to use the rectangular plate to do the experiment? More discussions are necessary.
We thank the Reviewer for the concern. The aim of the experiment was to obtain a set of data with noise induced by vibrations similar to the one that can be obtained from a rotating helicopter blade during flight in order to demonstrate the principle of the proposed data treatment approach. In this preliminary research, the authors did not pursue the goal to compare the vibrations with the real-life reference, which will be the aim of the subsequent studies. We clarified this in Section 4.1 (lines 357–360).

Reviewer 3 Report
The paper is interesting and reports the Noise Cancellation of Helicopter Blade Deformations Measurement by FBGs. Major revisions are needed as follows.
- In the sentence: "The main advantages of FBG-based techniques are low weight, immunity to electromagnetic interference, small dimensions (the optical fiber diameter is 125 μm), and absence of electrical power supply to the sensors. " it needs a reference for general application of FBGs. Please add: Toward commercial polymer fiber Bragg grating sensors: Review and applications, Journal of Lightwave Technology 37 (11), 2605-2615.
2. How the FBGs were fabricated? They were purchase? Please add this point.
3. Associated errors to the results need to be presented in the graphs.
4. A comparison with actual commercial sensors needs to be highlighted
Author Response
We would like to thank the Reviewer for appreciating the paper and the valuable remarks and suggestions.
The paper is interesting and reports the Noise Cancellation of Helicopter Blade Deformations Measurement by FBGs.
The authors thank the anonymous Reviewer for his/her appreciation.
1. In the sentence: "The main advantages of FBG-based techniques are low weight, immunity to electromagnetic interference, small dimensions (the optical fiber diameter is 125 μm), and absence of electrical power supply to the sensors. " it needs a reference for general application of FBGs. Please add: Toward commercial polymer fiber Bragg grating sensors: Review and applications, Journal of Lightwave Technology 37 (11), 2605-2615.
Thank you for the suggestion. The reference was added.
2. How the FBGs were fabricated? They were purchase? Please add this point.
Thank you for the question. The following information was added to the text (lines 348 – 350): “The FBGs used in the experiment were manufactured at the Department of Radiophotonics and Microwave Technologies of KNRTU-KAI (Kazan) using a common fabrication technique based on the Lloyd interferometer.”
3. Associated errors to the results need to be presented in the graphs.
Two treatment methods applied in this paper do not contain the treatment errors. All errors are associated with experimental equipment used. The error of the FBG central wavelength definition of the interrogation device used in the experimental setup was 1 pm.
4. A comparison with actual commercial sensors needs to be highlighted.
The sensors and the interrogation device used in this research are commercial and are already applied in commercial projects.

Round 2
Reviewer 1 Report
Thank you to the authors for taking into account all the comments and revise/modify the paragraphs accordingly. In my opinion, a great effort has been made by the authors and the paper can be published as presented when the following issue is addressed.
Figures 5 to 9 show wavelengths shifts detected by the FBG interrogation device with respect to “Data Points” or “Load (g)”.
X axes title names are correct since they refer to the information of the data. However, the Y axes titles, even if the authors have include the physical quantity measured (that is nanometers in all cases), they include an explanation of the data depicted in each figure in the axis title and result confusing. This explanation need to be included in the figure footnote and within the text to explain the results but not in the Y axis title. It is requested to change Y axes names to “Wavelength shift (nm)” since this is the physical quantity that has been measured in all this figures.
Author Response
The Y-axes titles were corrected accordingly. We thank the reviewer.
Reviewer 2 Report
The revisions and responses are satisfactory.
Author Response
We thank the reviewer.
Reviewer 3 Report
The authors have improved the paper as suggested.
Author Response
We thank the reviewer.